# Importance of attributes and willingness to pay for oral anticoagulant therapy in patients with atrial fibrillation in China: A discrete choice experiment

Jiaxi Zhao[1,2☯], Hao Wang[3☯], Xue Li[1,4,5], Yang Hu[4], Vincent K. C. Yan[1], Carlos K. H. Wong[1,6], Yutao Guo[7], Marco K. H. Cheung[4], Gregory Y. H. Lip[8,9], Chung-Wah Siu[10], Hung-Fat Tse[10,11,12], Esther W. Chan[1,2,5]*

1 Centre for Safe Medication Practice and Research, Department of Pharmacology and Pharmacy, Li Ka Shing Faculty of Medicine, The University of Hong Kong, Hong Kong SAR, China, 2 The University of Hong Kong Shenzhen Institute of Research and Innovation, Shenzhen, China, 3 Department of Cardiology, The Second Medical Center, National Clinical Research Center for Geriatric Diseases, Chinese PLA General Hospital, Beijing, China, 4 Department of Medicine, Li Ka Shing Faculty of Medicine, The University of Hong Kong, Hong Kong SAR, China, 5 Laboratory of Data Discovery for Health (D²4H), Hong Kong Science Park, New Territories, Hong Kong SAR, China, 6 Department of Family Medicine and Primary Care, Li Ka Shing Faculty of Medicine, The University of Hong Kong, Hong Kong SAR, China, 7 Department of Cardiology, Chinese PLA General Hospital, Beijing, China, 8 Liverpool Centre for Cardiovascular Science, University of Liverpool and Liverpool Heart & Chest Hospital, Liverpool, United Kingdom, 9 Department of Clinical Medicine, Aalborg University, Aalborg, Denmark, 10 Cardiology Division, Department of Medicine, Li Ka Shing Faculty of Medicine, The University of Hong Kong, Hong Kong SAR, China, 11 Hong Kong-Guangdong Stem Cell and Regenerative Medicine Research Centre, The University of Hong Kong and Guangzhou Institutes of Biomedicine and Health, Hong Kong SAR, China, 12 Department of Medicine, The University of Hong Kong-Shenzhen Hospital, Shenzhen, China

☯ These authors contributed equally to this work.

* ewchan@hku.hk

**Data Availability Statement:** As the data was collected via a questionnaire survey, due to the nature of ethical restriction in this study,

## Abstract

### Background

Adherence to oral anticoagulant therapy in patients with atrial fibrillation (AF) in China is low. Patient preference, one of the main reasons for discontinuation of oral anticoagulant therapy, is an unfamiliar concept in China.

### Methods and findings

A discrete choice experiment (DCE) was conducted to quantify patient preference on 7 attributes of oral anticoagulant therapy: antidote (yes/no), food–drug interaction (yes/no), frequency of blood monitoring (no need, every 6/3/1 month[s]), risk of nonfatal major bleeding (0.7/3.1/5.5/7.8[%]), risk of nonfatal stroke (ischemic/hemorrhagic) or systemic embolism (0.6/3.2/5.8/8.4[%]), risk of nonfatal acute myocardial infarction (AMI) (0.2/1.0/1.8/2.5[%]), and monthly out-of-pocket cost (0/120/240/360 RMB) (0 to 56 USD). A total of 16 scenarios were generated by using D-Efficient design and were randomly divided into 2 blocks. Eligible patients were recruited and interviewed from outpatient and inpatient settings of 2 public hospitals in Beijing and Shenzhen, respectively. Patients were presented with 8 scenarios

participants of this study did not agree for their un-anonymized data to be shared publicly. Authors of this article confirm that all anonymized data supporting the findings of this study are available within the article and its supplementary materials.

**Funding:** EWC received funding from National Natural Science Fundation of China for this study (Ref Number: 71704149). (URL is http://www.nsfc.gov.cn/) The funder had no role in study design, data collection and analysis, decision to publish or preparation of the manuscript.

**Competing interests:** The authors declare no competing interests.

**Abbreviations:** AF, atrial fibrillation; AMI, acute myocardial infarction; CHA2DS2-VASc, congestive heart failure (C), hypertension (H), age $\geq$ 75 (A2), diabetes mellitus (D), prior stroke or transient ischemic attack or thromboembolism (S2), vascular disease (V), age between 65 and 74 (A), sex category (Sc); CI, confident interval; DCE, discrete choice experiment; EHRA, European Heart Rhythm Association; INR, international normalized ratio; MI, myocardial infarction; mWTP, marginal willingness to pay; NOAC, non-vitamin K antagonist oral anticoagulant; RMB, Ren Min Bi (unit of Chinese currency); SD, standard deviation; USD, US dollar (unit of American currency); WTP, willingness to pay.

and asked to select 1 of 3 options: 2 unlabeled hypothetical treatments and 1 opt-out option. Mixed logit regression model was used for estimating patients' preferences of attributes of oral anticoagulants and willingness to pay (WTP) with adjustments for age, sex, education level, income level, city, self-evaluated health score, histories of cardiovascular disease/other vascular disease/any stroke/any bleeding, and use of anticoagulant/antiplatelet therapy. A total of 506 patients were recruited between May 2018 and December 2019 (mean age 70.3 years, 42.1% women). Patients were mainly concerned about the risks of AMI ($\beta$: −1.03; 95% CI: −1.31, −0.75; $p < 0.001$), stroke or systemic embolism ($\beta$: −0.81; 95% CI: −0.90, −0.73; $p < 0.001$), and major bleeding ($\beta$: −0.69; 95% CI: −0.78, −0.60; $p < 0.001$) and were willing to pay more, from up to 798 RMB to 536 RMB (124 to 83 USD) monthly. The least concerning attribute was frequency of blood monitoring ($\beta$: −0.31; 95% CI: −0.39, −0.24; $p < 0.001$). Patients had more concerns about food–drug interactions even exceeding preferences on the 3 risks, if they had a history of stroke or bleeding ($\beta$: −2.47; 95% CI: −3.92, −1.02; $p < 0.001$), recruited from Beijing ($\beta$: −1.82; 95% CI: −2.56, −1.07; $p < 0.001$), or men ($\beta$: −0.96; 95% CI: −1.36, −0.56; $p < 0.001$). Patients with lower educational attainment or lower income weighted all attributes lower, and their WTP for incremental efficacy and safety was minimal. Since the patients were recruited from 2 major hospitals from developed cities in China, further studies with better representative samples would be needed.

## Conclusions

Patients with AF in China were mainly concerned about the safety and effectiveness of oral anticoagulant therapy. The preference weighting on food–drug interaction varied widely. Patients with lower educational attainment or income levels and less experience of bleeding or stroke had more reservations about paying for oral anticoagulant therapies with superior efficacy, safety, and convenience of use.

## Author summary

### Why was this study done?

- Patient adherence with oral anticoagulant therapy in patients with atrial fibrillation (AF) in China is poor. Patient preference on treatment attributes and their willingness to pay (WTP) might be key factors in patient adherence.

- Results from studies using discrete choice experiment (DCE) outside China suggest that preferences and WTP varied in different populations.

- Quantitative evaluation of patient preference on attributes of oral anticoagulants and WTP in China remains an area that needs further exploration.

### What did the researchers do and find?

- We conducted a survey using a DCE design to investigate preferences on attributes of oral anticoagulants and WTP in patients with AF from Beijing and Shenzhen, 2 large representative cities in China.

- Patients were asked to choose their preferred treatment based on their evaluation of the attributes of oral anticoagulant: antidote, food–drug interaction, frequency of blood monitoring, risk of nonfatal major bleeding, risk of nonfatal stroke or systemic embolism, risk of nonfatal acute myocardial infarction (AMI), and monthly out-of-pocket cost.

- Patients were mostly concerned about the risks of AMI, stroke or systemic embolism, and major bleeding and were willing to pay 798 to 536 RMB (124 to 83 USD) per month to obtain a better drug with less risks. Food–drug interaction was weighted to be the most important attribute by patients who had history of bleeding or stroke or who were from Beijing.

### What do these findings mean?

- The relatively highest preference weights for the risk of both cardiovascular and cerebrovascular events indicate that patients might be more attentive to the safety of vital organs than the main side effects of anticoagulant therapy.

- Patients with different demographic or health characteristics have specific preferences. The relatively high preference weight and marginal willingness to pay (mWTP) for food–drug interaction among specific patients suggest that patients might prefer the most direct option offering rapid improvement.

## Introduction

Oral anticoagulant therapy is a standard management for stroke prevention in patients with atrial fibrillation (AF). In line with the increasing global prevalence of AF over the past decade [1], the age- and sex-standardized prevalence of AF in Chinese over 35 year olds has continued to rise, reaching 0.71% in 2017 [2]. Due to rapid action onset and offset, predictable pharmacodynamics, and fewer food-drug interactions, non-vitamin K antagonist oral anticoagulants (NOACs) are increasingly favored in clinical practice over warfarin worldwide [3] and has been recommended in the latest international clinical practice guidelines [3–7]. However, adherence to NOAC therapy among patients in China is lower than warfarin [8,9], although the use of NOACs was found to be increasing rapidly [10].

   One of the main reasons for discontinuation of NOACs is patient preference or affordability [8]. A multicenter cross-sectional study found that out-of-pocket payment was the strongest predictor for declining anticoagulant therapy use in China [11]. In China, although NOACs have 92% of the oral anticoagulant therapy market share in terms of monetary value, only 28% of patients with AF were prescribed a NOAC in 2017 [10]. Currently, for NOACs prescribed in outpatient setting, only 70% of the cost will be subsidized [11]. Achieving a sufficient reduction in the risk of ischemic stroke with an acceptable elevation in bleeding risk is an important clinical trade-off that patients need to consider [12]. In addition, patients' perspectives concerning difficulties of access to the healthcare service and the complexity of treatment will also influence medication adherence [13,14], a crucial issue for anticoagulant therapy as emphasized in the 2018 European Heart Rhythm Association (EHRA) practical guide [6].

However, current evidence is insufficient to rank the importance of factors affecting patients' medication adherence in China.

Discrete choice experiment (DCE) is a method of eliciting and quantifying preferences and exploring trade-offs between the attributes (characteristics) of alternatives [15]. An Australian DCE study [12] indicated a consensus that patients were likely to choose more effective and safer oral anticoagulant therapies and would prefer to pay more for a reduction in the risk of stroke and bleeding. Although one United States study [16] supported this viewpoint, the lower weight of these 2 attributes compared to blood testing and dietary restrictions heightened the concern for preference heterogeneity between different populations. This heterogeneity existed in the same country. Another US study [17] revealed that patients with cardiovascular disease were not as concerned about the effectiveness and safety, including fatal events, of oral anticoagulants as patients in the previous 2 studies.

Inconsistent results, as well as cultural differences between China and high-income countries, which could potentially impact the perspectives of patients receiving treatment, mean that conclusions from previous studies conducted outside China may not be generalizable to Chinese patients. To our knowledge, this is the first study to apply DCE to the investigation of preferences on attributes of oral anticoagulant therapy and willingness to pay (WTP) among patients with AF in China.

## Methods

A cross-sectional questionnaire based on a face-to-face survey was conducted. Eligible patients were recruited from both outpatient and inpatient settings of the cardiology departments of 2 public hospitals, the Chinese People's Liberation Army General Hospital and The University of Hong Kong—Shenzhen Hospital, located in 2 representative cities, Beijing and Shenzhen, respectively. There was no prospective protocol published for this study.

### Discrete choice experiment

Understanding the impact of the crucial attributes of treatments on patients' decision-making is more informative than merely understanding their preference of the actual product as a whole. Attributes may have different weights that influence patients' evaluation of the trade-offs between treatments. Although traditional surveys can be used to obtain quantitative preferences about products as a whole, it is difficult to compare the relative importance of these individual attributes. DCE is a mature method to easily measure the level of risk/benefit that patients would take/give up for one attribute in exchange for the benefit of another. By using this approach, we may be able to ascertain the dominating factors that influence patients when choosing treatments, as well as the price they are prepared to spend to be free of or obtain specific attributes. Therefore, the results could assist clinicians, pharmaceutical companies, governments, and other stakeholders in adjusting treatment plan, improving products, and offering more reasonable pricing, which could improve medication adherence. This study used a DCE design to quantitatively assess patients' preferences of the most important attributes for oral anticoagulants. The entire experiment adhered to the recommendations of the International Society for Pharmacoeconomics and Outcome Research protocol for conducting DCE [18].

### Target population and sample size

A convenience sampling approach was used to recruit patients who were (i) diagnosed with AF; (ii) aged $\geq 18$ years old; (iii) with congestive heart failure (C), hypertension (H), age $\geq 75$ ($A_2$), diabetes mellitus (D), prior stroke or transient ischemic attack or thromboembolism ($S_2$),

vascular disease (V), age between 65 and 74 (A), sex category (Sc) (CHA$_2$DS$_2$-VASc) $\geq 1$ for men and CHA$_2$DS$_2$-VASc $\geq 2$ for women; and (iv) without communication impediments.

## Attribute and level identification

A total of 7 attributes of oral anticoagulants (Table 1) were identified based on the evidence from previous studies [12,16,17,19–21], in addition to the ranking by an expert panel with 40 cardiologists and 40 eligible patients. Two levels were assigned to the attributes of antidote and food–drug interaction. Patients who use warfarin drugs are subject to monthly blood monitoring although this is not necessary for NOAC users. We used 4 levels, every 1/3/6 month(s) or not required/necessary, for this attribute in order to determine the preferred monitoring frequency for clinical application. Four levels were used for the risk of nonfatal major bleeding (0.7/3.1/5.5/7.8[%]), the risk of nonfatal stroke or systemic embolism (0.6/3.2/5.8/8.4[%]), and the risk of nonfatal acute myocardial infarction (AMI) (0.2/1.0/1.8/2.5[%]) [22–27]. The highest and lowest rates of the aforementioned 3 risk events when using oral anticoagulants

**Table 1. Selected attributes of oral anticoagulants.**

| Attributes* | Levels |
|---|---|
| Antidote could be provided if there was a side effect when administrating oral anticoagulant | Yes/no |
| Food–drug interaction exists when administrating oral anticoagulant | Yes/no |
| Frequency of visiting hospital to monitor coagulation function | No need for monitoring<br>Every 6 months<br>Every 3 months<br>Every 1 month |
| Probability that you would encounter a nonfatal major bleeding event when administrating oral anticoagulant | 0.7/3.1/5.5/7.8 (%) |
| Probability that you would encounter a nonfatal stroke (either ischemic or hemorrhagic) or systemic embolism event when administrating oral anticoagulant | 0.6/3.2/5.8/8.4 (%) |
| Probability that you would encounter a nonfatal AMI event when administrating oral anticoagulant | 0.2/1.0/1.8/2.5 (%) |
| Out-of-pocket monthly cost | 0/120/240/360 (RMB)# |

*Bleeding is the main side effect for stroke prevention in the treatment using oral anticoagulants among patients with AF. Attributes of oral anticoagulants valued by patients the most would logically be the risks of bleeding and stroke, which represent the safety and effectiveness of drugs. Some characteristics, which are not direct attributes of oral anticoagulants such as risk of MI, may still concerned by patients in China. In addition, any attributes that are crucial and vary between different oral anticoagulants may also influence patients' choice of anticoagulation therapy.

Risks of bleeding, stroke, and MI were expanded into different levels based on the severity. They were fatal bleeding risk, nonfatal major bleeding risk, nonfatal minor bleeding risk, nonfatal any bleeding risk, fatal stroke or MI risk, nonfatal major stroke risk, nonfatal minor stroke risk, nonfatal any stroke risk, and nonfatal MI risk. Besides, antidote, blood test requirement, food–drug interaction, dose adjustment, frequency of drug administration, as well as out-of-pocket cost, which is always the key attribute for mWTP calculation, were ranked by 40 cardiologists and 40 eligible patients.

Risks of bleeding, stroke, and MI were the most concerned attributes of oral anticoagulant for both patients and clinicians. In order to avoid potential bias induced by fatal risks, which dominated patients' decision in our exploratory experiment, fatal risks were removed. As patients had difficulties in distinguishing ischemic stroke and hemorrhagic stroke in our exploratory experiment, stroke were described as ischemic or hemorrhagic.

#1 USD $\approx$ 6.4 RMB.

RMB indicates Ren Min Bi, the unit of Chinese currency.

AF, atrial fibrillation; AMI, acute myocardial infarction; MI, myocardial infarction; mWTP, marginal willingness to pay.

reported from observational studies were used as the ceiling and baseline levels of attributes, respectively, with the ranges equally divided into 4. The monthly out-of-pocket cost of NOAC for a patient with medical insurance is approximately 150 to 300 RMB (23 to 47 USD). As most patients would have medical insurance, the levels for this attribute were expressed as 0/120/240/360 RMB.

## DCE design

To prevent an impractically large sample size reaching sufficient power, we used a D-Efficient design to generate a set of choices consisting of 16 combinations from a full factorial design, which had $2 \times 2 \times 4 \times 4 \times 4 \times 4 \times 4 = 4,096$ hypothetical profiles for each alternative in this study. In order to give a manageable workload to each patient while harnessing the statistical properties, 16 scenarios with different combinations of hypothetical profiles of alternatives were randomly divided into 2 blocks, with each block including 8 scenarios. Eligible patients were randomly allocated to one of the 2 blocks.

The names of alternatives could distract patients' attention from the targeting attributes and bias their preferences, especially when the population has a different understanding about the products [28,29]. Therefore, patients were asked to choose one preferred oral anticoagulant from 2 unlabeled hypothetical treatments. One "opt-out" option (i.e., no anticoagulant therapy) would be chosen if patients disliked either treatments based on the provided attribute levels. This could provide a more realistic preference [30].

A testing scenario with a dominated alternative was included at the beginning to assess patients' understanding (S1 File). Investigators would immediately explain the meaning of levels of attributes to those who did not choose the dominated alternative. Those patients were allowed to continue answering the questionnaire only when their understanding were confirmed by investigators. DCEs were generated by Ngene V1.1.2 (ChoiceMetrics, W201/599 Pacific Highway, St Leonards, Sydney, NSW 2065, AU) (S2 and S3 Files).

## Sample size

According to Orme's rule of thumb formula, minimum sample size = 

$$\frac{500 * (largest\ level\ of\ attributes)}{(number\ of\ scenarios) * (number\ of\ non\ opt-out\ alternatives)}$$ [31]. Given the conditions of 2 alternatives (excluding opt-out), and maximum 4 levels for attributes, as well as 8 scenarios presented to each patient to obtain the largest minimum required sample size, at least 125 participants in total are required. To ensure sufficient power, we aimed to recruit 300 patients for each site [31].

## Data collection

Eligible patients identified between May 2018 and December 2019 by cardiologists were transferred to trained investigators for the subsequent investigation. The purpose of this study, basic information about oral anticoagulant therapy, and how to complete the DCE questionnaire were explained by the site investigator. All patients who signed the consent forms were included in the analyses. Face-to-face interviews were conducted via paper-based questionnaires (S4 File). Patient demographic characteristics, such as age, sex, occupation, education level, monthly income after tax, medical insurance type, history of cardiovascular medication, history of diseases, self-evaluated health score, were also collected for analyses (Table 2).

## Analyses

Mixed logit regression model was used for the main analysis to estimate patients' preferences on each attribute. Considering the assumption of independence for outcomes in mixed logit

**Table 2. Characteristics of recruited patients.**

| Characteristics | Total (*n* = 506) | Beijing (*n* = 299) | Shenzhen (*n* = 207)* |
|---|---|---|---|
| **Number of patients who failed in the testing scenario**[#] | 36 (7.1) | 20 (6.7) | 16 (7.7) |
| **Sociodemographic profile** | | | |
| Age, mean (SD), y | 70.3 (12.7) | 70.8 (13.2) | 69.5 (12.0) |
| Women | 213 (42.1) | 99 (33.1) | 114 (55.1) |
| Inpatient setting | 366 (72.3) | 201 (67.2) | 165 (79.7) |
| Education | | | |
| Illiteracy | 19 (3.8) | 2 (0.7) | 17 (8.2) |
| Primary school | 71 (14.0) | 36 (12.0) | 35 (16.9) |
| Secondary school | 197 (38.9) | 107 (35.8) | 90 (43.5) |
| Undergraduate | 193 (38.1) | 134 (44.8) | 59 (28.5) |
| Postgraduate | 26 (5.1) | 20 (6.7) | 6 (2.9) |
| Occupation | | | |
| Full-time job | 74 (14.6) | 48 (16.1) | 26 (12.6) |
| Part-time job | 1 (0.2) | 0 (0) | 1 (0.5) |
| Student | 0 (0) | 0 (0) | 0 (0) |
| Peasant | 53 (10.5) | 39 (13.0) | 14 (6.8) |
| Housewife | 28 (5.5) | 17 (5.7) | 11 (5.3) |
| Retired | 344 (68.0) | 195 (65.2) | 149 (72.0) |
| Unemployed | 6 (1.2) | 0 (0) | 6 (2.9) |
| Individual monthly income after tax (RMB) | | | |
| ≤5,000 | 215 (42.5) | 91 (30.4) | 124 (60.0) |
| 5,001–12,000 | 277 (54.7) | 201 (67.2) | 76 (36.7) |
| >12,000 | 14 (2.8) | 7 (2.3) | 7 (3.4) |
| **Medical condition** | | | |
| Self-evaluated health score, mean (SD) | 62.4 (17.4) | 60.2 (16.9) | 65.7 (17.5) |
| Self-reported histories of diseases | | | |
| Cardiovascular disease | 391 (77.3) | 220 (73.6) | 171 (82.6) |
| Other vascular disease | 235 (46.4) | 195 (65.2) | 40 (19.3) |
| Stroke/transient ischemic attack/systemic embolism | 117 (23.1) | 88 (29.4) | 29 (14.0) |
| Renal impairment | 70 (13.8) | 36 (12.0) | 34 (16.4) |
| Bleeding (major/minor) | 53 (10.5) | 38 (12.7) | 15 (7.2) |
| Liver impairment | 20 (4.0) | 17 (5.7) | 3 (1.4) |
| Other thrombosis | 11 (2.2) | 11 (3.7) | 0 (0) |
| Self-reported histories of medication use | | | |
| Anticoagulant | 394 (77.9) | 214 (71.6) | 180 (87.0) |
| Antiarrhythmic agent | 318 (62.8) | 148 (49.5) | 170 (82.1) |
| Lipid-lowering agent | 311 (61.5) | 177 (59.2) | 134 (64.7) |
| Antihypertensive agent | 319 (63.0) | 183 (61.2) | 136 (65.7) |
| Antihyperglycemic agent | 117 (23.1) | 65 (21.7) | 52 (25.1) |
| Antianginal agent | 104 (20.6) | 93 (31.1) | 11 (5.3) |
| Antiplatelet agent | 95 (18.8) | 62 (20.7) | 33 (15.9) |
| Type of medical insurance[&] | | | |
| Employee insurance | 171 (33.8) | 97 (32.4) | 74 (35.7) |
| Resident insurance | 90 (17.8) | 7 (2.3) | 83 (40.1) |
| Military medical service | 63 (12.5) | 62 (20.7) | 1 (0.5) |
| Rural insurance | 60 (11.9) | 38 (12.7) | 22 (10.6) |
| Free medical service | 44 (8.7) | 40 (13.4) | 4 (1.9) |

(*Continued*)

**Table 2.** (Continued)

| Characteristics | Total (*n* = 506) | Beijing (*n* = 299) | Shenzhen (*n* = 207)* |
|---|---|---|---|
| Retirement insurance | 34 (6.7) | 26 (8.7) | 8 (3.9) |
| Commercial insurance | 14 (2.8) | 11 (3.7) | 3 (1.4) |
| None | 28 (5.5) | 21 (7.0) | 7 (3.4) |

*Patients recruitment in Shenzhen site was terminated due to the COVID-19 pandemic in early 2020.

#A testing scenario (S1 File) was designed to test whether patients understood the DCE questionnaire. In this scenario, drug A is obviously worse than drug B, because drug A has higher risks of suffering from side effect, including bleeding, MI, has higher risk of suffering from stroke, has food–drug interaction but no antidote, and has a higher price. If patients did not choose drug B, it indicated they might not understand this questionnaire. Investigators would immediately explain the meaning of levels of attributes to them. These patients were allowed to answer the questionnaire only when their understandings were confirmed by investigators. After investigators' explanation, these patients were considered to have similar understanding to other patients and were included in analyses.

&Employee insurance indicates mandatory insurance for those who are in employment; resident insurance indicates mandatory insurance for those living in the city. Military medical service indicates free medical services for those who are working or were previously in the army; rural insurance indicates insurance for people who living in rural areas; free medical services are for those meet specific criteria; retirement insurance applies to those already retired; and commercial insurance is applicable to all.

RMB indicates Ren Min Bi, the unit of Chinese currency.

COVID-19, Coronavirus Disease 2019; DCE, discrete choice experiment; MI, myocardial infarction.

regression model would have been violated because the selection of "opt-out" option would be highly dependent on the selection of "drug A" and "drug B", scenarios with opt-out selected were removed from the main analysis. Antidote and food–drug interaction were treated as binary; the frequency of blood monitoring was treated as ordinal; all of the risk attributes and the monthly out-of-pocket cost were treated as continuous variables. Factors that could influence patients' understandings for questionnaire, such as education level, income level, and city, were adjusted. Factors that could represent patients' previous experiences on anticoagulant therapy, such as self-evaluated health score, histories of cardiovascular disease/other vascular disease/any stroke/any bleeding, and use of anticoagulant/antiplatelet, were adjusted. As common confounders, age and sex were also adjusted. Random effects for each attribute were taken into consideration to minimize the effects from residual confounders that induced heterogeneity in patients, except for out-of-pocket cost, which was treated as a fixed effect to calculate WTP. The correlation between any pair of attributes was also adjusted in the model. McFadden Pseudo $R^2$ was used to evaluate the goodness of fit for models [32]. Marginal willingness to pay (mWTP) was calculated by assessing the ratio of the preference of other attributes to the preference of out-of-pocket cost (Table 3). Subgroup analyses by sex, education level (under or above high school), income level (under or above 5,000 RMB [775 USD] per month), with or without history of any stroke/bleeding, and cities were performed (Table 4).

Several sensitivity analyses were performed to test the robustness of our results. Firstly, patients who failed the testing scenario were removed from the analysis to assess the quality of our data (S5 File). Furthermore, to determine more specific preferences, the frequency of blood monitoring was treated as a categorical variable (S6 File). Block of scenarios was adjusted in mixed logit regression model to assess the potential impact of blocks on preferences (S7 File). An analysis stratified by outpatient and inpatient setting was conducted to assess the potential influence from different settings (S8 File). As medical insurance may influence patients' preferences, an analysis with the exclusion of all patients who did not have any medical insurance was performed (S9 File). Two additional statistical models, nested logit model and multinomial logit model, were used to assess the influence of the opt-out option (S10 and S11 Files).

**Table 3. Preference weights estimated by mixed logit regression model and mWTP (RMB\*) in main analysis.**

| Attribute | Crude estimation | | Adjusted estimation[#] | | mWTP (95% CI) |
|---|---|---|---|---|---|
| | β (95% CI) | *p*-Value[&] | β (95% CI) | *p*-Value[&] | |
| Out-of-pocket monthly cost | −0.0011 (−0.0014, −0.0008) | <0.001 | −0.0013 (−0.0016, −0.0009) | <0.001 | - |
| Risk of nonfatal AMI | −0.85 (−1.08, −0.62) | <0.001 | −1.03 (−1.31, −0.75) | <0.001 | −798 (−1,247, −518) |
| Risk of nonfatal stroke or systemic embolism | −0.70 (−0.76, −0.64) | <0.001 | −0.81 (−0.90, −0.73) | <0.001 | −632 (−880, −488) |
| Risk of nonfatal major bleeding | −0.58 (−0.65, −0.52) | <0.001 | −0.69 (−0.78, −0.60) | <0.001 | −536 (−762, −403) |
| Food–drug interaction | −0.36 (−0.55, −0.16) | <0.001 | −0.63 (−0.88, −0.38) | <0.001 | −489 (−764, −288) |
| Antidote | 0.45 (0.22, 0.68) | <0.001 | 0.43 (0.17, 0.69) | 0.001 | 338 (125, 639) |
| Frequency of blood monitoring | −0.28 (−0.34, −0.22) | <0.001 | −0.31 (−0.39, −0.24) | <0.001 | −244 (−347, −180) |
| Model specifications | Crude: Log likelihood = −2,081; McFadden Pseudo $R^2$ = 0.1898 | | | | |
| | Adjusted: Log likelihood = −2,030; McFadden Pseudo $R^2$ = 0.2096 | | | | |

\*1 USD ≈ 6.4 RMB.

[#]Adjusted by age, sex, education level, income level, city, self-evaluated health score, history of cardiovascular disease/other vascular disease/any stroke/any bleeding, and use of anticoagulant/antiplatelet; the correlation between any pair of attributes was also involved in the model.

[&]*p*-Values for coefficients were obtained by Wald test.

mWTP indicates marginal willingness to pay; it was calculated under adjusted estimations. Krinsky and Robb's method was used to calculate 95% CI of mWTP. RMB indicates Ren Min Bi, the unit of Chinese currency. β indicates coefficient and represents relative weight; negative value indicates negative preference.

AMI, acute myocardial infarction.

The main analysis, subgroup analyses, and sensitivity analyses (to test the robustness of different models) were prespecified, conducted, and reported as planned (S10 and S11 Files). Additional sensitivity analyses (S5–S9 Files) that had been added to test the robustness of the results against the main results were data driven. All statistical analyses were conducted using R 3.6.3 (R Core Team 2020, Austria). Patient characteristics were summarized as mean (SD) for continuous variables and in frequencies (percentage) for categorical variables. All hypothesis tests were 2 sided, with a significance level of 0.05. This study was approved by the ethics committees of both Chinese People's Liberation Army General Hospital and The University of Hong Kong-Shenzhen Hospital.

## Results

### Characteristics of patients (Table 2)

After excluding one patient who did not answer any question, a total of 506 patients (299 from Beijing and 207 from Shenzhen), with mean age 70.3 years, of which 42.1% were women, were successfully recruited and completed consent forms. Regardless of testing scenario, each patient was given 8 scenarios with 2 treatment alternatives and one opt-out option to choose from, and a result of 12,144 (506 × 8 × 3) completed records were expected. After removing 201 empty responses, 11,943 records were kept for further analyses. The first and second blocks were answered by 254 and 252 patients, respectively (S4 File). A total of 36 out of 506 patients (7.1%) did not choose the dominant drug in the testing scenario.

### Main analysis (Table 3)

The positive and negative values of coefficients represent patients' preference and dislike for the corresponding attribute, respectively. The greater the absolute value of the coefficient, the more weight for changes per unit of that attribute were taken into consideration when patients evaluated the trade-offs.

**Table 4. Preference weights estimated by mixed logit regression model and mWTP (RMB\*) in subgroup analyses.**

**Stratified by sex**

| Attribute | Men (n = 293) | | | | | Women (n = 213) | | | | |
|---|---|---|---|---|---|---|---|---|---|---|
| | Crude estimation | | Adjusted estimation[#] | | mWTP (95% CI) | Crude estimation | | Adjusted estimation[#] | | mWTP (95% CI) |
| | β (95% CI) | p-Value[&] | β (95% CI) | p-Value[&] | | β (95% CI) | p-Value[&] | β (95% CI) | p-Value[&] | |
| Out-of-pocket monthly cost | −0.0008 (−0.0013, −0.0003) | <0.001 | −0.0012 (−0.0018, −0.0006) | <0.001 | - | −0.0017 (−0.0023, −0.0012) | <0.001 | −0.0019 (−0.0025, −0.0014) | <0.001 | - |
| Risk of nonfatal AMI | −0.91 (−1.25, −0.58) | <0.001 | −1.11 (−1.55, −0.67) | <0.001 | −932 (−2,043, −473) | −0.76 (−1.11, −0.41) | <0.001 | −0.83 (−1.28, −0.38) | <0.001 | −425 (−777, −183) |
| Risk of nonfatal stroke or systemic embolism | −0.80 (−0.90, −0.71) | <0.001 | −1.05 (−1.21, −0.88) | <0.001 | −882 (−1,673, −597) | −0.61 (−0.71, −0.52) | <0.001 | −0.70 (−0.83, −0.56) | <0.001 | −359 (−524, −262) |
| Risk of nonfatal major bleeding | −0.65 (−0.75, −0.56) | <0.001 | −0.87 (−1.03, −0.71) | <0.001 | −731 (−1,423, −479) | −0.55 (−0.65, −0.46) | <0.001 | −0.62 (−0.76, −0.48) | <0.001 | −317 (−478, −220) |
| Food–drug interaction | −0.45 (−0.73, −0.17) | 0.002 | −0.96 (−1.36, −0.56) | <0.001 | −805 (−1,521, −458) | −0.24 (−0.55, 0.07) | 0.12 | −0.34 (−0.73, 0.05) | 0.09 | −174 (−396, 22) |
| Antidote | 0.60 (0.28, 0.93) | <0.001 | 0.44 (0.04, 0.84) | 0.03 | 368 (27, 1,010) | 0.27 (−0.08, 0.63) | 0.13 | 0.34 (−0.10, 0.77) | 0.13 | 172 (−47, 445) |
| Frequency of blood monitoring | −0.30 (−0.38, −0.22) | <0.001 | −0.38 (−0.50, −0.26) | <0.001 | −320 (−615, −198) | −0.26 (−0.35, −0.17) | <0.001 | −0.33 (−0.45, −0.22) | <0.001 | −171 (−256, −110) |
| Model specification | Men (crude model): Log likelihood = −1,160; McFadden Pseudo R² = 0.2114 | | | | | | | | | |
| | Men (adjusted model): Log likelihood = −1,114; McFadden Pseudo R² = 0.2424 | | | | | | | | | |
| | Women (crude model): Log likelihood = −903; McFadden Pseudo R² = 0.1771 | | | | | | | | | |
| | Women (adjusted model): Log likelihood = −873; McFadden Pseudo R² = 0.2051 | | | | | | | | | |

**Stratified by education level**

| Attribute | At or under high school (n = 287) | | | | | Over high school (n = 219) | | | | |
|---|---|---|---|---|---|---|---|---|---|---|
| | Crude estimation | | Adjusted estimation[#] | | mWTP (95% CI) | Crude estimation | | Adjusted estimation[#] | | mWTP (95% CI) |
| | β (95% CI) | p-Value[&] | β (95% CI) | p-Value[&] | | β (95% CI) | p-Value[&] | β (95% CI) | p-Value[&] | |
| Out-of-pocket monthly cost | −0.0023 (−0.0028, −0.0018) | <0.001 | −0.0025 (−0.0031, −0.0020) | <0.001 | - | 0.0004 (−0.0002, 0.0010) | 0.17 | 0.0003 (−0.0004, 0.0010) | 0.37 | - |
| Risk of nonfatal AMI | −0.87 (−1.18, −0.57) | <0.001 | −1.05 (−1.41, −0.68) | <0.001 | −413 (−626, −250) | −1.00 (−1.45, −0.54) | <0.001 | −1.28 (−1.91, −0.65) | <0.001 | 4,210 (−39,680, 39,239) |
| Risk of nonfatal stroke or systemic embolism | −0.62 (−0.70, −0.54) | <0.001 | −0.71 (−0.82, −0.61) | <0.001 | −280 (−357, −225) | −0.98 (−1.12, −0.83) | <0.001 | −1.20 (−1.44, −0.97) | <0.001 | 3,955 (−38,180, 37,774) |
| Risk of nonfatal major bleeding | −0.51 (−0.59, −0.42) | <0.001 | −0.60 (−0.71, −0.48) | <0.001 | −235 (−310, −179) | −0.85 (−0.99, −0.71) | <0.001 | −1.05 (−1.28, −0.82) | <0.001 | 3,459 (−32,474, 33,325) |
| Food–drug interaction | −0.27 (−0.53, 0.00) | 0.05 | −0.54 (−0.89, −0.20) | 0.002 | −215 (−362, −81) | −0.58 (−0.96, −0.21) | 0.002 | −0.92 (−1.44, −0.39) | <0.001 | 3,021 (−30,866, 29,776) |
| Antidote | 0.29 (−0.04, 0.61) | 0.08 | 0.23 (−0.14, 0.61) | 0.23 | 92 (−57, 264) | 0.86 (0.45, 1.26) | <0.001 | 1.09 (0.53, 1.66) | <0.001 | −3,604 (−33,435, 33,248) |
| Frequency of blood monitoring | −0.24 (−0.32, −0.16) | <0.001 | −0.28 (−0.37, −0.18) | <0.001 | −109 (−150, −73) | −0.38 (−0.48, −0.27) | <0.001 | −0.46 (−0.60, −0.32) | <0.001 | 1,511 (−14,489, 14,201) |

(*Continued*)

**Table 4.** (*Continued*)

| Model specification | At or under high school (crude model): Log likelihood = −1,201; McFadden Pseudo $R^2$ = 0.1902 |
|---|---|
| | At or under high school (adjusted model): Log likelihood = −1,168; McFadden Pseudo $R^2$ = 0.2128 |
| | Over high school (crude model): Log likelihood = −792; McFadden Pseudo $R^2$ = 0.2695 |
| | Over high school (adjusted model): Log likelihood = −764; McFadden Pseudo $R^2$ = 0.2956 |

**Stratified by income level[§]**

| Attribute | ≤5,000 RMB monthly income (n = 215) | | | | | >5,000 RMB monthly income (n = 291) | | | | |
|---|---|---|---|---|---|---|---|---|---|---|
| | Crude estimation | | Adjusted estimation[#] | | mWTP (95% CI) | Crude estimation | | Adjusted estimation[#] | | mWTP (95% CI) |
| | β (95% CI) | p-Value[&] | β (95% CI) | p-Value[&] | | β (95% CI) | p-Value[&] | β (95% CI) | p-Value[&] | |
| Out-of-pocket monthly cost | −0.0028 (−0.0033, −0.0022) | <0.001 | −0.0032 (−0.0039, −0.0026) | <0.001 | - | 0.0003 (−0.0002, 0.0008) | 0.30 | 0.0004 (−0.0020, 0.0011) | 0.15 | - |
| Risk of nonfatal AMI | −0.64 (−0.98, −0.31) | <0.001 | −0.77 (−1.21, −0.33) | <0.001 | −241 (−408, −98) | −1.21 (−1.59, −0.82) | <0.001 | −1.50 (−1.98, −1.01) | <0.001 | 3,325 (−21,086, 29,843) |
| Risk of nonfatal stroke or systemic embolism | −0.50 (−0.59, −0.42) | <0.001 | −0.63 (−0.75, −0.50) | <0.001 | −194 (−251, −149) | −1.00 (−1.11, −0.88) | <0.001 | −1.13 (−1.31, −0.97) | <0.001 | 2,521 (−17,112, 22,781) |
| Risk of nonfatal major bleeding | −0.41 (−0.50, −0.32) | <0.001 | −0.52 (−0.66, −0.39) | <0.001 | −162 (−220, −116) | −0.85 (−0.96, −0.73) | <0.001 | −0.98 (−1.15, −0.81) | <0.001 | 2,169 (−14,246, 19,240) |
| Food–drug interaction | −0.20 (−0.50, 0.09) | 0.17 | −0.48 (−0.87, −0.08) | 0.02 | −148 (−276, −25) | −0.44 (−0.76, −0.11) | 0.008 | −0.63 (−1.05, −0.22) | 0.003 | 1,408 (−10,195, 14,345) |
| Antidote | 0.32 (−0.04, 0.68) | 0.09 | 0.24 (−0.19, 0.68) | 0.28 | 75 (−61, 222) | 0.77 (0.43, 1.11) | <0.001 | 0.81 (0.37, 1.25) | <0.001 | −1,802 (−16,330, 10,988) |
| Frequency of blood monitoring | −0.29 (−0.38, −0.20) | <0.001 | −0.33 (−0.45, −0.22) | <0.001 | −103 (−141, −69) | −0.34 (−0.43, −0.25) | <0.001 | −0.41 (−0.53, −0.29) | <0.001 | 905 (−6,471, 8,079) |
| Model specification | ≤5,000 RMB monthly income (crude model): Log likelihood = −927; McFadden Pseudo $R^2$ = 0.1632 | | | | | | | | | |
| | ≤5,000 RMB monthly income (adjusted model): Log likelihood = −896; McFadden Pseudo $R^2$ = 0.1915 | | | | | | | | | |
| | >5,000 RMB monthly income (crude model): Log likelihood = −1,061; McFadden Pseudo $R^2$ = 0.2736 | | | | | | | | | |
| | >5,000 RMB monthly income (adjusted model): Log likelihood = −1,022; McFadden Pseudo $R^2$ = 0.2998 | | | | | | | | | |

**Stratified by history of bleeding or stroke**

| Attribute | Without history of bleeding or stroke (n = 354) | | | | | With history of bleeding or stroke (n = 152) | | | | |
|---|---|---|---|---|---|---|---|---|---|---|
| | Crude estimation | | Adjusted estimation[#] | | mWTP (95% CI) | Crude estimation | | Adjusted estimation[#] | | mWTP (95% CI) |
| | β (95% CI) | p-Value[&] | β (95% CI) | p-Value[&] | | β (95% CI) | p-Value[&] | β (95% CI) | p-Value[&] | |
| Out-of-pocket monthly cost | −0.0012 (−0.0015, −0.0008) | <0.001 | −0.0017 (−0.0022, −0.0013) | <0.001 | - | −0.0007 (−0.0014, 0.0001) | 0.10 | −0.0018 (−0.0029, 0.0007) | 0.001 | - |
| Risk of nonfatal AMI | −0.86 (−1.09, −0.62) | <0.001 | −0.79 (−1.14, −0.45) | <0.001 | −457 (−750, −245) | −1.55 (−2.21, −0.90) | <0.001 | −2.30 (−4.14, −0.47) | 0.01 | −1,279 (−4,044, −219) |
| Risk of nonfatal stroke or systemic embolism | −0.69 (−0.76, −0.63) | <0.001 | −0.74 (−0.85, −0.62) | <0.001 | −423 (−576, −325) | −1.29 (−1.52, −1.07) | <0.001 | −2.03 (−2.69, −1.38) | <0.001 | −1,129 (−2,681, −646) |
| Risk of nonfatal major bleeding | −0.57 (−0.64, −0.51) | <0.001 | −0.62 (−0.74, −0.51) | <0.001 | −358 (−501, −266) | −0.99 (−1.19, −0.80) | <0.001 | −1.60 (−2.25, −0.94) | <0.001 | −886 (−2,265, −434) |
| Food–drug interaction | −0.35 (−0.55, −0.15) | <0.001 | −0.51 (−0.81, −0.21) | <0.001 | −294 (−486, −123) | −0.90 (−1.41, −0.39) | <0.001 | −2.47 (−3.92, −1.02) | <0.001 | −1,372 (−3,588, −528) |
| Antidote | 0.44 (0.21, 0.68) | <0.001 | 0.31 (−0.01, 0.62) | 0.06 | 177 (−5, 400) | 1.02 (0.33, 1.70) | 0.004 | 1.11 (−0.48, 2.69) | 0.17 | 615 (−258, 2,455) |
| Frequency of blood monitoring | −0.28 (−0.34, −0.22) | <0.001 | −0.34 (−0.42, −0.25) | <0.001 | −194 (−271, −141) | −0.27 (−0.40, −0.14) | <0.001 | −0.43 (−0.70, −0.15) | 0.002 | −236 (−628, −79) |

(*Continued*)

**Table 4.** (Continued)

| Model specification | |
|---|---|
| | Without history of bleeding or stroke (crude model): Log likelihood = −2,010; McFadden Pseudo $R^2$ = 0.1869 |
| | Without history of bleeding or stroke (adjusted model): Log likelihood = −1,446; McFadden Pseudo $R^2$ = 0.1879 |
| | With history of bleeding or stroke (crude model): Log likelihood = −542; McFadden Pseudo $R^2$ = 0.3120 |
| | With history of bleeding or stroke (adjusted model): Log likelihood = −510; McFadden Pseudo $R^2$ = 0.3521 |

**Stratified by city**

| Attribute | Beijing (n = 299) | | | | | Shenzhen (n = 207) | | | | |
|---|---|---|---|---|---|---|---|---|---|---|
| | Crude estimation | | Adjusted estimation[#] | | mWTP (95% CI) | Crude estimation | | Adjusted estimation[#] | | mWTP (95% CI) |
| | β (95% CI) | p-Value[&] | β (95% CI) | p-Value[&] | | β (95% CI) | p-Value[&] | β (95% CI) | p-Value[&] | |
| Out-of-pocket monthly cost | −0.0020 (−0.0026, −0.0014) | <0.001 | −0.0034 (−0.0045, −0.0024) | <0.001 | - | −0.0009 (−0.0014, −0.0003) | 0.003 | −0.0011 (−0.0019, −0.0004) | 0.004 | - |
| Risk of nonfatal AMI | −1.19 (−1.60, −0.78) | <0.001 | −1.56 (−2.44, −0.68) | <0.001 | −455 (−836, −183) | −0.50 (−0.86, −0.13) | 0.007 | −0.79 (−1.37, −0.22) | 0.007 | −711 (−2,619, −162) |
| Risk of nonfatal stroke or systemic embolism | −0.99 (−1.11, −0.86) | <0.001 | −1.67 (−2.02, −1.33) | <0.001 | −488 (−674, −373) | −0.57 (−0.68, −0.47) | <0.001 | −0.87 (−1.10, −0.65) | <0.001 | −783 (−2,339, −458) |
| Risk of nonfatal major bleeding | −0.78 (−0.90, −0.67) | <0.001 | −1.31 (−1.63, −0.99) | <0.001 | −383 (−553, −273) | −0.52 (−0.62, −0.41) | <0.001 | −0.79 (−1.01, −0.56) | <0.001 | −705 (−2,118, −395) |
| Food–drug interaction | −0.64 (−0.98, −0.31) | <0.001 | −1.82 (−2.56, −1.07) | <0.001 | −531 (−803, −321) | −0.004 (−0.33, 0.32) | 0.98 | −0.19 (−0.74, 0.36) | 0.49 | −172 (−808, 518) |
| Antidote | 0.82 (0.35, 1.28) | <0.001 | 0.56 (−0.28, 1.40) | 0.19 | 163 (−85, 455) | 0.14 (−0.20, 0.48) | 0.42 | 0.20 (−0.35, 0.75) | 0.47 | 180 (−322, 1,240) |
| Frequency of blood monitoring | −0.15 (−0.25, −0.06) | 0.001 | −0.34 (−0.53, −0.15) | <0.001 | −100 (−161, −47) | −0.43 (−0.53, −0.32) | <0.001 | −0.68 (−0.88, −0.49) | <0.001 | −613 (−1,771, −357) |
| Model specification | Beijing (crude model): Log likelihood = −1,119; McFadden Pseudo $R^2$ = 0.2721 | | | | | | | | | |
| | Beijing (adjusted model): Log likelihood = −1,044; McFadden Pseudo $R^2$ = 0.3210 | | | | | | | | | |
| | Shenzhen (crude model): Log likelihood = −869; McFadden Pseudo $R^2$ = 0.1562 | | | | | | | | | |
| | Shenzhen (adjusted model): Log likelihood = −833; McFadden Pseudo $R^2$ = 0.1917 | | | | | | | | | |

[*]1 USD ≈ 6.4 RMB.

[#]Adjusted by age, sex, education level, income level, city, self-evaluated health score, history of cardiovascular disease/other vascular disease/any stroke/any bleeding, and use of anticoagulant/antiplatelet; the correlation between any pair of attributes was also involved in the model.

[&]p-Values for coefficients were obtained by Wald test.

[§]Using 5,000 RMB as an individual monthly income cutoff point as this is the current threshold for personal income tax payment in China.

mWTP indicates marginal willingness to pay; it was calculated under adjusted estimations. Krinsky and Robb's method was used to calculate 95% CI of mWTP. RMB indicates Ren Min Bi, the unit of Chinese currency. β indicates coefficient and represents relative weight; negative value indicates negative preference.

AMI, acute myocardial infarction.

Compared to other attributes, the relatively higher weights for the risk of nonfatal AMI (β: −1.03; 95% CI: −1.31, −0.75; $p < 0.001$), the risk of nonfatal stroke or systemic embolism (β: −0.81; 95% CI: −0.90, −0.73; $p < 0.001$), and the risk of nonfatal major bleeding (β: −0.69; 95% CI: −0.78, −0.60; $p < 0.001$) indicate that the issues patients had the most concern about during anticoagulant therapy were safety and effectiveness. The impact of the frequency of blood monitoring on decision-making was the weakest (β: −0.31; 95% CI: −0.39, −0.24; $p < 0.001$), and patients disliked frequent hospital visits. Patients preferred treatment with the antidote (β: 0.43; 95% CI: 0.17, 0.69; $p = 0.001$), while avoiding food–drug interactions (β: −0.63; 95% CI: −0.88, −0.38; $p < 0.001$).

Patients were willing to pay most of the cost for oral anticoagulant therapy: 798 RMB (β: −798; 95% CI: −1,247, −518), 632 RMB (β: −632; 95% CI: −880, −488), and 536 RMB (β: −532; 95% CI: −762, −403), respectively, to reduce 1% of the risk of nonfatal AMI, nonfatal stroke or systemic embolism, and nonfatal major bleeding. They began to hesitate about paying more to

be free of food–drug interaction (β: −489; 95% CI: −764, −288) or to obtain the extra antidote for side effects and reactions (β: 338; 95% CI: 125, 639) and were willing to pay only the minimum to reduce the frequency of blood monitoring (β: −244; 95% CI: −347, −180).

## Subgroup analyses (Table 4)

Men were slightly more concerned about almost all of the attributes compared to women and were willing to pay double that of women. The impact of food–drug interaction on men (β: −0.96; 95% CI: −1.36, −0.56; $p < 0.001$) was much stronger than that on women (β: −0.34; 95% CI: −0.73, 0.05; $p = 0.09$), and they were willing to pay over 4 times more than women (805 versus 174 RMB [125 versus 27 USD]).

As is widely known, education level correlates with income level. The similarity in patterns of preferences on all attributes could also be observed for both subgroups; patients with lower educational attainment (at or under high school) or who had a lower income (≤5,000 RMB/month) were less concerned about all of the attributes and were only willing to pay the minimum to obtain extra benefits or get rid of disadvantages. mWTP varied widely among patients with higher levels of education and income.

History of any bleeding or stroke influenced patients' preference. Anyone with a history valued all attributes at the same high weights and were willing to pay up to 5 times of those without a history. For this group, food–drug interaction was the most concerning attribute (β: −2.47; 95% CI: −3.92, −1.02; $p < 0.001$).

We stratified the analysis by cities to test heterogeneity. Compared to Shenzhen, the effect size of all attributes on patients from Beijing was chiefly doubled. The food–drug interaction had the strongest impact on patients from Beijing (β: −1.82; 95% CI: −2.56, −1.07; $p < 0.001$). However, they were only willing to pay less than the amount that patients from Shenzhen were prepared to pay for most of the attributes.

## Sensitivity analyses (S5–S11 Files)

First, we removed patients who did not initially select the dominated alternative in the testing scenario to assess data quality. Preferences and mWTP for each attribute were consistent with the primary analysis (S5 File).

With consideration that the linear assumption might be violated in the ordinal variable frequency of blood monitoring since the intervals between each level were unequal, we also treated this as a categorical variable (S6 File). Compared to nonnecessity for monitoring, patients preferred less frequent 6 monthly hospital visits (β: 0.43; 95% CI: 0.18, 0.69; $p < 0.001$). The extent of dislike deteriorated as the frequency increased (every 3 months β: −0.30; 95% CI: −0.54, −0.05; $p = 0.02$ and every 1 month β: −0.85; 95% CI: −1.09, −0.61; $p < 0.001$).

The consistent results of the sensitivity analyses, which adjusted blocks of scenarios, stratified by outpatient and inpatient settings, and excluded patients who did not have any medical insurance, from main analysis, indicated that there would not be obvious influence from those aforementioned factors on patients' preferences (S7–S9 Files).

By adopting a nested logit model to assess the effects of opt-out on decision-making, the preferences for all attributes were principally similar to that in the main analysis. However, due to the inability to deal with the correlated alternative selection, the results from multinomial logistic regression were not consistent with the main analysis (S10 and S11 Files).

## Discussion

In view of the underutilization of oral anticoagulant therapy and poor medication adherence among patients with AF in China, this study provides an in-depth exploration of the crucial

factors that could affect patients' preferences, allowing either possible product improvements or therapeutic focused adjustments. In addition, as apixaban and edoxaban have not yet been approved for use in China, the results from mWTP could assist with reasonable pricing for the remaining NOACs in China.

The magnitude of concerns about the attributes vary in different populations. In general, the risks of AMI, stroke or systemic embolism, and major bleeding, which could intuitively reflect drug safety and effectiveness, emerged as characteristics for which patients were willing to pay most for improvement. The relatively higher preference weights for the risk of both cardiovascular and cerebrovascular events indicate that patients might be more attentive to the safety of vital organs than the common severe side effects during anticoagulant therapy. Disease conditions of the heart or brain necessitate inevitable long-term medical care, which place a heavy burden on most families in China. This finding is consistent with another US study [17] that was of relative importance for AMI and stroke compared to major bleeding in our study ($\frac{\beta_{AMI}}{\beta_{Bleed}} = 1.49$, $\frac{\beta_{Stroke}}{\beta_{Bleed}} = 1.17$) and is similar to theirs ($\frac{\beta_{AMI}}{\beta_{Bleed}} = 2.87$, $\frac{\beta_{Stroke}}{\beta_{Bleed}} = 2.26$). However, the relative importance of blood monitoring compared to major bleeding in our study ($\frac{\beta_{Monitoring}}{\beta_{Bleed}} = 0.45$) was 9 to 15 times higher than that in 2 studies ($\frac{\beta_{Monitoring}}{\beta_{Bleed}} = 0.049$) [12] ($\frac{\beta_{Monitoring}}{\beta_{Bleed}} = 0.032$) [17], which suggested that Chinese patients might have different perspectives about the attributes compared to patients from high-income countries. A recent study found that the general spatial accessibility to hierarchical healthcare facilities in Shenzhen is unevenly distributed and concentrated [33]. Also, the supply of healthcare resources at primary care facilities is far from sufficient. This might reveal the reason why patients in our study were more concerned about blood monitoring. More frequent blood test monitoring is inconvenient for many patients who do not reside near their primary healthcare facility. Another study focused on oral anticoagulation adherence of patients in China suggested that consecutive international normalized ratio (INR) monitoring was a major concern of patients, especially for those who are older, women, and have previous stroke history, as they are at a higher risk of bleeding and need more intensive INR monitoring [34].

Patients with different demographic or health characteristics have their specific preferences. The most surprising finding was that the preference weight of food–drug interaction was the highest among men, patients who had been recruited from Beijing, and who had a history of bleeding or stroke. The dietary habits of Chinese is complex. Green vegetables, rich in vitamin K, which could influence the effectiveness of vitamin K antagonist oral anticoagulants, are usually a main dietary preference. Drinking alcohol is also taboo while under drug treatment. However, one of the main social activities in China is attending banquets, especially common among northern residents and men. This might be the reason that the preference weight among these patients was high. Patients with history of bleeding or stroke who were concerned mainly with food–drug interactions could have experienced severe adverse reactions caused by poor dietary habits. In addition, the relatively high mWTP for this attribute suggest that patients might prefer the most direct option offering rapid improvement.

Currently, patients with AF in China are generally required to pay an additional 150 to 300 RMB per month for anticoagulant therapy using NOAC. According to our results, patients were willing to pay a little more than the current monthly expenditure for oral anticoagulants with better safety, effectiveness, and convenience profiles. Nevertheless, if the drug attribute preferences of patients with low income were the same as that of the overall population, their WTP would be understandably low. It is worth noting that the preferences of patients with high income and high educational attainment varied widely. The insignificance of out-of-

pocket costs suggests that current incremental cost was within the affordable range for this population.

## Limitations

There were several limitations in this study. First, as discussed in other stated preference surveys, the true preferences were not revealed because the decisions made were merely hypothetical. Second, all scenarios only considered a limited number of attributes. Other attributes, such as the risk of bleeding with different severity levels and side effect related death, for example, may also reflect other preferences. Nevertheless, those 7 included attributes were the one mostly concerned by patients in other DCE studies. Regardless of sample size increase, too many attributes would also increase patient workload, affecting their decision-making due to information overload and reduce the reliability of the result. Some attributes, such as death, could be a distraction and bias the result. Third, we cannot fully ascertain whether patients prefer NOACs or warfarin due to the unlabeled design. However, patients might be biased by the names of alternatives without an understanding of the products. Moreover, the unlabeled approach may help identify patients' preferences more accurately. Forth, due to the nature of DCE, there may not be a flawless analytic approach to obtain a comprehensive result. However, we have conducted several sensitivity analyses using different models with consistent results. Fifth, the levels of risks of major bleeding, stroke and systemic embolism, and myocardial infarction were obtained from studies mainly on non-Chinese/Asian populations. The estimated preferences on these attributes might be less generalizable to a Chinese population. However, the findings may not be too biased due to the linear assumption, as long as the differences of those risks between the 2 populations are not large and are still useful especially for the firsthand evidence from an exploratory study. Lastly, the 2 hospitals that were used to recruit patients are the major hospitals in 2 developed and affluent cities in China. Patients from these 2 hospitals may not be representative of patients in China in general.

## Conclusions

Patients with AF in China were mainly concerned with the safety and effectiveness of oral anticoagulant therapy. The preference weight on food–drug interaction varied widely. Patients with lower educational attainment or low income and less experience of bleeding or stroke were more conservative about paying for oral anticoagulant therapies with superior efficacy, safety, and convenience of use.

### Ethical approval and informed consent

This study was approved by Medical Ethics Committees of the Chinese People's Liberation Army General Hospital (Reference number: S2018-043-01) and HKU-Shenzhen Hospital (Reference number: [2018]78).

### Supporting information

**S1 File. Sample for scenario 1 (for the purpose of testing patients' understanding).** (DOCX)

**S2 File. Summary of the D-Efficient design using Ngene.** (DOCX)

**S3 File. Results of balance for alternatives in designed questionnaire.** (DOCX)

**S4 File. Patients' responses.**
(DOCX)

**S5 File. Preference weights estimated by mixed logit regression model excluding patients who failed in the test scenario ($n$ = 470).**
(DOCX)

**S6 File. Preference weights estimated by mixed logit regression model with frequency of blood monitoring as a categorical variable ($n$ = 506).**
(DOCX)

**S7 File. Preference weights estimated by mixed logit regression model with blocks of scenarios being included for adjustment ($n$ = 506).**
(DOCX)

**S8 File. Preference weights estimated by mixed logit regression model stratified by outpatient and inpatient setting ($n$ = 504).**
(DOCX)

**S9 File. Preference weights estimated by mixed logit regression model with the exclusion of 28 patients who did not have any medical insurance ($n$ = 478).**
(DOCX)

**S10 File. Preference weights estimated by nested logit model with opt-out option included ($n$ = 506).**
(DOCX)

**S11 File. Preference weights estimated by multinomial logit model with opt-out option included ($n$ = 506).**
(DOCX)

## Acknowledgments

We thank Dr. Kelvin K.H. Yiu and his colleagues from the Department of Cardiology, The University of Hong Kong Shenzhen Hospital for their assistance in coordinating the patient recruitment process. We thank Ms. Yue Wei, Ms. Le Gao, Ms. Lam Lam, and Ms. Ying Shi, who helped with data entry.

We acknowledge that EWC has received honorarium from the Hospital Authority, research grants from Narcotics Division of the Security Bureau of HKSAR, National Health and Medical Research Council (NHMRC, Australia), National Natural Science Foundation of China (NSFC), Research Fund Secretariat of the Food and Health Bureau (HMRF, HKSAR), Research Grants Council (RGC, HKSAR), Wellcome Trust; Amgen, AstraZeneca, Bayer, Bristol-Myers Squibb, Janssen, Pfizer, RGA, and Takeda outside the submitted work. XL has received the research grant from Health and Medical Research Fund, Food and Health Bureau of Hong Kong Government; research and educational grants from Janssen and Pfizer; internal funds from the University of Hong Kong; consultancy fees, unrelated to this work. CKHW has received research funding from the EuroQoL Group Research Foundation, the Hong Kong Research Grants Council, and the Hong Kong Health and Medical Research Fund, Food and Health Bureau. GYHL is a consultant and speaker for BMS/Pfizer, Boehringer Ingelheim, and Daiichi-Sankyo; no fees are received personally. Aforementioned funders had no role in study design, data collection and analysis, decision to publish, or preparation of the manuscript.

## Author Contributions

**Conceptualization:** Jiaxi Zhao, Hao Wang, Xue Li, Carlos K. H. Wong, Yutao Guo, Gregory Y. H. Lip, Esther W. Chan.

**Data curation:** Jiaxi Zhao, Hao Wang, Yang Hu, Yutao Guo.

**Formal analysis:** Jiaxi Zhao, Hao Wang, Xue Li, Vincent K. C. Yan, Carlos K. H. Wong, Marco K. H. Cheung, Esther W. Chan.

**Funding acquisition:** Hao Wang, Xue Li, Carlos K. H. Wong, Yutao Guo, Esther W. Chan.

**Investigation:** Jiaxi Zhao, Hao Wang, Xue Li, Yang Hu, Yutao Guo, Gregory Y. H. Lip, Esther W. Chan.

**Methodology:** Jiaxi Zhao, Hao Wang, Xue Li, Carlos K. H. Wong, Gregory Y. H. Lip, Esther W. Chan.

**Project administration:** Jiaxi Zhao, Esther W. Chan.

**Resources:** Yutao Guo, Chung-Wah Siu, Hung-Fat Tse, Esther W. Chan.

**Supervision:** Yutao Guo, Gregory Y. H. Lip, Chung-Wah Siu, Hung-Fat Tse, Esther W. Chan.

**Validation:** Xue Li, Gregory Y. H. Lip, Esther W. Chan.

**Writing – original draft:** Jiaxi Zhao, Hao Wang, Xue Li, Esther W. Chan.

**Writing – review & editing:** Jiaxi Zhao, Hao Wang, Xue Li, Yang Hu, Vincent K. C. Yan, Carlos K. H. Wong, Yutao Guo, Marco K. H. Cheung, Gregory Y. H. Lip, Chung-Wah Siu, Hung-Fat Tse, Esther W. Chan.

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
