## [Editor Report · Decision Letter 0]

17 Feb 2021

Dear Dr Chan, 

Thank you for submitting your manuscript entitled "Understanding oral anticoagulant preferences of patients with atrial fibrillation in China and willingness-to-pay: Evidence from a discrete choice experiment" for consideration by PLOS Medicine.

Your manuscript has now been evaluated by the PLOS Medicine editorial staff as well as by an academic editor with relevant expertise and I am writing to let you know that we would like to send your submission out for external peer review.

Kind regards,

Dr Raffaella Bosurgi

Executive Editor 

PLOS Medicine

---

## [Decision Letter · Decision Letter 1]

29 Apr 2021

Dear Dr. Chan,

Thank you very much for submitting your manuscript "Understanding oral anticoagulant preferences of patients with atrial fibrillation in China and willingness-to-pay: Evidence from a discrete choice experiment" (PMEDICINE-D-21-00677R1) for consideration at PLOS Medicine. 

[LINK]

In light of these reviews, I am afraid that we will not be able to accept the manuscript for publication in the journal in its current form, but we would like to consider a revised version that addresses the reviewers' and editors' comments. Obviously we cannot make any decision about publication until we have seen the revised manuscript and your response, and we plan to seek re-review by one or more of the reviewers. 

We expect to receive your revised manuscript by May 20 2021 11:59PM. Please email us (plosmedicine@plos.org) if you have any questions or concerns.

We look forward to receiving your revised manuscript. 

Sincerely,

Beryne Odeny, 

PLOS Medicine

plosmedicine.org

• Please revise your title according to PLOS Medicine's style. Your title must be nondeclarative and not a question. It should begin with main concept if possible. Please place the study design ("A discrete choice experiment,") in the subtitle (i.e., after a colon). 

• The Data Availability Statement (DAS) requires revision. For each data source used in your study: 

a) If the data are owned by a third party but freely available upon request, please note this and state the owner of the data set and contact information for data requests (web or email address). Note that a study author cannot be the contact person for the data.

b) If the data are not freely available, please include an appropriate contact (web or email address) for inquiries (again, this cannot be a study author).

• In the abstract Methods and Findings:

- Please structure your abstract using the PLOS Medicine headings (Background, Methods and Findings, Conclusions). Please combine the Methods and Findings sections into one section, “Methods and findings”.

-Please ensure that all numbers presented in the abstract are present and identical to numbers presented in the main manuscript text.

- Please quantify the main results with both 95% CIs and p values.

- Please include the important dependent variables that are adjusted for in the analyses.

- In the last sentence of the Abstract Methods and Findings section, please describe the main limitation(s) of the study's methodology.

• Did your study have a prospective protocol or analysis plan? Please state this (either way) early in the Methods section.

• In statistical methods, please refer to any post-hoc corrections to correct for multiple comparisons during your statistical analyses. If these were not performed please justify the reasons. Please refer to our statistical reporting guidelines for assistance (https://journals.plos.org/plosone/s/submission-guidelines.#loc-statistical-reporting)

• Please describe how you selected your adjustment variables. 

• In the Methods and Results section:

- Please provide both adjusted and unadjusted estimates. In all tables and supplements, please provide the unadjusted comparisons for the adjusted comparisons

- Please provide 95% CIs and p values for estimates. Include in both tables and main text

- When a p value is given, please specify the statistical test used to determine it.

• The term "trend" is used to refer to a nonsignificant P value. The term trend should be used only when the test for trend has been conducted. Please revise accordingly.

• The terms gender and sex are not interchangeable (as discussed in http://www.who.int/gender/whatisgender/en/ ); please use the appropriate term.

• Please refer to high income countries rather than "Western" countries.

• Please replace “warfarin users” with “patients who use warfarin drugs.”

• Please use the "Vancouver" style for reference formatting and see our website for other reference guidelines https://journals.plos.org/plosmedicine/s/submission-guidelines#loc-references.

• Please define the abbreviations in tables and/or Figures e.g. AMI, RMB

Please include line numbers in your next draft.

Comments from the reviewers:

Reviewer #1: Understanding oral anticoagulant preferences of patients with AF in China and willingness-to-pay

This is a well-written report of a rigorously conducted conjoint analysis study of patient choice factors related to OACs in China. longitudinal study of student learning about "EBM" in a pharmacy school. Questions and suggestions whose responses could strengthen the manuscript are provided below. Relatively minor grammatical copyediting is required. 

Introduction

Well-written and justified.

Methods

Choice of the 7 attributes: It is stated these were chosen based on literature search and "ranking by an expert panel with forty cardiologists and forty eligible patients", but the supplement doesn't disclose the list from which the 7 were chosen. More disclosure about this seems necessary since readers may wonder why some attributes were or were not chosen. For example, "risk of non-fatal acute myocardial infarction" is questionable as a risk of NOAC treatment. This was initially identified as a possible risk with dabigatran, but largely clarified subsequently as likely not a drug-attributable risk. "Risk of non-fatal stroke (either ischemic or hemorrhagic)" contains a dichotomy... one is an efficacy attribute of the drugs, and one is a toxicity attribute. There is an inherent trade-off between ischemic and hemorrhagic stroke between OACs and it is misleading to include both in a single attribute. Justify. 

Profiles/scenarios were used instead of individual-attribute tradeoffs for sample-size and participant burden reasons, and "names of alternatives could distract patients' attention from the targeting attributes, and bias their preferences" were justifiable performed. However, this implies that the scenarios corresponded to attributes of actual drugs. If so, the Results and Discussion should "unmask" these so more meaningful discussion about the actual therapeutic implications of the results is possible. If not, why does the Limitations section say, "we do not know whether patients prefer NOACs or warfarin due to the unlabelled design". 

Results

Well-written. No specific comments. 

Discussion

The authors do a good job of interpreting the results in the context of Chinese culture. 

"...we included the 7 most important attributes on the basis of the literature review and pilot study" - see above comments and juitfiy or modify this statement. 

Conclusion

"...better treatment." Clarify what is meant by "better" here. More precision is required. 

Reviewer #2: This is an interesting and useful study on the oral anticoagulant preferences of patients with atrial fibrillation in China using discrete choice experiment (DCE) design. However, there are a few major issues needing attention.

1) Study design. There seems to be some inconsistencies between study design, conduct and analysis. The DCE design includes 7 attributes with up to 4 levels which leads to 4096 hypothetical profiles for each alternative which is a lot. It said in the methods "16 scenarios were randomly divided into two blocks with 8 for each. Eligible patients were also randomly assigned to two blocks". It's very confusing as all the sample size and analyses later on were based on 8 scenarios. Should stick to 16 or 8 scenarios if designed so, but designed 16 scenarios then reduced to 8 by randomised into tow blocks? Why not started with 8 or merge the 16 into 8? Any bias introduced? Any adjustment in the analyses? Because the patients in different blocks will have different 8 scenarios and then how come to analyse them together?

2) Again, the DCE was designed with 16 scenarios and 3 alternatives. But both the sample size and main analyses were based on 2 alternatives (logit regression) by exculding the 'opt-out' option. Why? and what's the point of designing 3 alternatives?

3) Sample size. Sample size was based on 2 alternatives (excluding opt-out) and 8 scenarios and maximum 4 levels for attributes, but actually it's 3 alternatives and 16 scenarios. Does it mean the current sample size of around 500 patients is under powered? 

4) Irrational respondent. It said in the method "Those who did not choose the dominated one were considered as irrational and were corrected by investigators before answering the remaining questions". It's inadequate to 'correct' irrational respondents but instead should identify and remove those irrational respondents. In table 1, need to report the both rational and irrational respondents within Beijing and Shenzhen columns.

5) Analysis. The main analyses were currently based on all patients. However, the irrational respondents should be removed, and the main analysis should focus on those rational respondents.

6) Again, I am not comfortable with the main analyses only on two alternatives by excluding 'opt-out', then why did authors design 3 alternatives with opt-out in the first place? The sensitivity analysis using multinomial

logistic regression by including 'opt-out' gave very different results to the current analyses was a bit worrying and needs further consideration and discussion on the study design and suitability of the main analyses on only two alternatives.

7) Supplementary Table 1. Attributes and levels should appear in the main text as Table 1.

8) Generalisability of the finding. The two hosptials in China, the Chinese PLA General Hospital and HKU-Shenzhen Hospital, are top hopspitals in China in two developed and affluent cities. Patients from these two hospital may not be representative of Chinese patients in general so need to discuss this in the limitation.

Reviewer #3: General Comments

This study is generally well-designed. The quality of survey design, analysis, and the manuscript is good enough. Whether this study is sufficiently innovative (originality) needs a third-party evaluation. 

Potential edits for the entire manuscript: This is preference and WTP on specific attributes of oral anticoagulants, rather than the preference on specific anticoagulants. Consider clarification through the entire manuscript. 

Intro

Paragraph 1

"weighted prevalence" is unclear terminology. What are the factors that the prevalence calculation was weighted for?

Paragraph 2

Cost may or may not be considered as a preference factor. As a reviewer, I would love to classify this as an external determinant rather than preference. Are we able to replace this with "affordability" ?

This sentence is unclear: "Although NOACs captured 92% of the oral anticoagulant therapy market, only 28% of patients with AF in China were covered in 2017." Does this sentence mean that a 92% of anticoagulation management involves NOAC in China, or across the world, but the usage rate becomes 28% when the analysis is limited to the Chinese AF population?

This sentence is unclear: "30% of costs are derived from patients attending outpatient departments". Does this mean that 30% of the AF management cost is incurred at outpatient setting including outpatient pharmacy and clinic visits, and 70% came from inpatient management?

Methods:

Patient recruitment from both outpatient and inpatient - see if there is a difference in the preference determined by the healthcare setting where patients were enrolled. 

Introduction for DCE seems two long. Probably you do not need the first two sentences of the "Discrete choice Experiment" section. 

Discrete Choice Experiment

This is unclear : "If patients were unable to complete the investigation, their guardians were considered as a proxy." Does this mean that care provider/guardian consented to participate and completed the DCE survey? IF that is the case, probably the whole analysis needs to be stratified by the type of respondent - patients vs. care providers/guardians - or needs to exclude responses from caregivers/guardians. 

Attribute and level identification.

Minor comment: cite reference to every single number makes the manuscript less easy to follow. Probably cite at the end of the sentence. 

This probably is self evident, but want to make sure that "antidote" means there is an antidote for the case of adverse reaction ?

Probably a short description on health literacy to understand the DCE questions would help. 

Results

Subgroup analysis:

If the authors can project the 95% confidence interval of the difference in WTP, that would improve the message. 

Discussion

the relative importance of blood monitoring compared to major bleeding in our study was 9-15 times higher than that in two western studies: Is this because of the access to healthcare facility? Or Shouldn't this be a matter?

It is too much self-evident from the data that, like the author describes, Chinese patients might have different perspectives about the attributes compared to western patients. There needs some description of where this perspective came from. 

This is not understood. Failed to find a linkage between "holistic manner" and "even weight".: Chinese patients might consider all aspects of oral anticoagulant therapy in a more holistic manner and thus weight attributes for oral anticoagulants more evenly. 

"The insignificance of out-of-pocket costs suggests that this population might be less concerned about the cost of treatment" might be better to be replaced by "The insignificance of out-of-pocket costs suggests that the current incremental cost is within the acceptable range of affordability". Preference would be influenced by changing out of pocket from the DCE, which was not tested. 

Reviewer #4: This interesting study applied discrete-choice-experiment methods to investigate preferences and willingness to pay of Chinese patients with atrial fibrillation for oral anticoagulation therapy. Results indicate that patients were mostly concerned about the risks of acute myocardial infarction, stroke or systemic embolism and major bleeding, but were willing to pay a better drug with less risks if they could easily afford it.

Major comments:

- Methods: 

* The risks of major bleeding, stroke/SE and myocardial infarction used as levels of the attributes, were based on 6 studies (references 23-28). However, were these studies based on outcome data in Chinese/Asian populations or not, given that risk estimates of other populations may be less generalizable to a Chinese population? Please discuss in the limitations.

* How many patients were invited in order to include 506 patients in total?

* Patients were randomly divided into two blocks with 8 scenarios for each. How was the randomisation process performed exactly? 

* 37 patients did not choose the dominated one in the testing scenario and were considered as irrational. They were corrected by investigators before answering the remaining questions. What does this mean? Were they educated on the meaning of the attributes? Please elaborate.

- Results:

* In the methods, it is mentioned that most patients would have medical insurance, which is important as the authors mention that the out-of-pocket cost of NOACs for a patient with medical insurance is approximately 150-300 RMB (23-47 USD). Based on table 1, 28 patients were included without medical insurance. Did this patient subgroup influence the results? 

* Scenarios with opt-out selected were removed from the main analysis. However, what were the driving attributes for these patients to not select an anticoagulant option? 

- Author summary:

* The current author summary is too large and the first two segments are almost identical to the abstract. I would recommend to summarize this section more and paraphrase duplicate statements.

Minor comments:

* In the first paragraph of the introduction, the term 'drug/food interactions' is used, whereas in the rest of the manuscript, the term 'food-drug interactions' is used. I would recommend to use the same term throughout the manuscript. 

- Grammatical/typing errors:

* Methods: "McFadden's Pseudo-R2" instead of "McFadden's Psuedo-R2"

* Discussion: "In addition, as apixaban and edoxaban have not yet been approved for use in China…" instead of "In addition, as apixaban andedoxaban have not yet been approved for use in China…"

[LINK]

---

## [Decision Letter · Decision Letter 2]

23 Jun 2021

Dear Dr. Chan,

Thank you very much for re-submitting your manuscript "Importance of attributes and willingness-to-pay for oral anticoagulant therapy in patients with atrial fibrillation in China: A discrete choice experiment" (PMEDICINE-D-21-00677R2) for review by PLOS Medicine.

I have discussed the paper with my colleagues and the academic editor and it was also seen again by three reviewers. I am pleased to say that provided the remaining editorial and production issues are dealt with we are planning to accept the paper for publication in the journal.

[LINK]

We look forward to receiving the revised manuscript by Jun 30 2021 11:59PM.   

Sincerely,

Beryne Odeny, 

Associate Editor 

PLOS Medicine

plosmedicine.org

Requests from Editors:

Thank your responses. Please address, in sufficient detail, the following outstanding points before we proceed:

1. In the manuscript text, please indicate: (1) that the analysis was prespecified and the specific hypotheses you intended to test, (2) the analytical methods by which you planned to test them, (3) the analyses you actually performed, and (4) when reported analyses differ from those that were planned, transparent explanations for differences that affect the reliability of the study's results. If a reported analysis was performed based on an interesting but unanticipated pattern in the data, please be clear that the analysis was data-driven.

2. Please define the all abbreviations in footnote for tables and/or Figures e.g. AMI, RMB1, mWTP and so forth.

Comments from Reviewers:

Reviewer #2: Many thanks authors for their great effort to improve the manuscript. All my questions/comments were professionally addressed. I am satisfied with the response and revision. No further issues needing attention. 

Reviewer #3: All my inquiries and comments on the previous version have been answered, or authors justified their approach if they could not provide readers with further data. Thank you. 

Reviewer #4: Thanks to the authors for revising the manuscript and responding my questions, I have no further comments.

[LINK]

---

## [Editor Report · Decision Letter 3]

12 Jul 2021

Dear Dr Chan, 

On behalf of my colleagues and the Academic Editor, Dr. Suzanne C. Cannegieter, I am pleased to inform you that we have agreed to publish your manuscript "Importance of attributes and willingness-to-pay for oral anticoagulant therapy in patients with atrial fibrillation in China: A discrete choice experiment" (PMEDICINE-D-21-00677R3) in PLOS Medicine.

PRESS

Sincerely, 

Beryne Odeny 

Associate Editor 

PLOS Medicine